# Fault tolerance in distributed systems using deep learning approaches

**Basem Assiri** [ID] ⍟, **Abdullah Sheneamer** [ID] ⍟ *

Computer Science Department, Faculty of Engineering and Computer Science, Jazan University, Jazan, Saudi Arabia

⍟ These authors contributed equally to this work.
* asheneamer@jazanu.edu.sa

**Data Availability Statement:** We use some standardized datasets and they are cited.

**Funding:** The authors gratefully acknowledge the funding of the Deanship of Graduate Studies and Scientific Research, Jazan University, Saudi Arabia, through Project Number: GSSRD-24.

## Abstract

Recently, distributed systems have become the backbone of technological development. It serves as the foundation for new trends technologies such as blockchain, the internet of things and others. A distributed system provides fault tolerance and decentralization, where a fault in any component does not result in a whole system failure. In addition, deep learning model enables processing data to find patterns, which helps in classification, regression, prediction, and clustering. This work employs deep learning to handle faults within distributed systems in three scenarios. Firstly, a faulty processor may not be able to produce the right output. Therefore, deep learning model uses the inputs and outputs of other processors to find patterns and produces the proper output of the faulty processor. Secondly, if a faulty possessor corrupts its inputs as well, then the deep learning model learns from the inputs and the outputs of successful processors and produces the proper output of the faulty processor, even with corrupted inputs. Thirdly, for unrelated data, in which the patterns of the input of the faulty processors differ from the patterns of the inputs of successful ones. In this case, the model is able to discover the new pattern and to be labeled as *unknown*. In the experiments, we use deep learning models like VGG16, VGG19, AlexNet LSTM and ResNet34, to investigate the performance of the deep learning in the three mentioned scenarios. For unstructured datasets, the accuracy of the models is affected by the size of the faulty data. The accuracy of all models lies between 60% when the size of the faulty data is 90%, and 96%, when the size of the faulty data is 90%. The structured datasets are not significantly affected by the portion of the faulty data and the accuracy reaches 99%.

## 1 Introduction

In the last decades, parallel and distributed computing has enhanced the advancement of technology. It serves as the foundation for new trends technologies such as blockchain technology, cloud storage, the internet of things and others. The concept of distributed systems is defined as a collection of components such as processors, storage, communicating networks, input tools, output tools, and actuators [1–3]. The components collaborate together, in a transparent way to appear as a single system, helping to achieve common goals. The distributed systems

**Competing interests:** The authors have declared that no competing interests exist.

provide many advantages such as decentralization, efficiency, high throughput, scalability, and reliability [1, 4, 5]. It achieves such advantages through distributing the workload over multiple processors and through fault tolerance.

Fault tolerance means that if any processor fails, the system can keep working. In decentralized distributed systems, there is no single point of failure, since processes run on multiple processors that are connected over a network and it can recover the faulty processor through redistributing its workload to other processors [6, 7] Actually, in distributed systems, there are many fault tolerance techniques [8, 9], as list below:

- Replication: it means to contain multiple copies of data and to store them in multiple places within the system. Therefore, when one copy is corrupted or not accessible because of a faulty processor, other copies are still correct and accessible. Moreover, the corrupt copy of data can be recovered and the system continues to work [9, 10].

- Redundancy: it allows to duplicate processing where the same process is duplicated and processed by more than one processor for more reliability and accuracy of the results, which enhances system functionality [9, 11, 12].

- Checkpoints: it enables periodically saving the system state for easy recovery in case of faults or even failure [13].

- System logs: they are used to timely record all events in the system which helps to track back in case of faults, errors, and intrusions.

- Load balancing and scheduling: load balancing allows takes' distribution according to the processors' capabilities to avoid processors' overloading and failure.

- Consensus protocols: they ensure that the majority of processors agree on some decisions. This helps to determine the faulty processors and data [14–16]

On the other hand, deep learning models help process large amounts of data to find patterns. This enables deep learning algorithms to perform classification, regression, prediction, and clustering. Actually, there are different categories of machine learning models [17, 18], as follow:

- Supervised learning: in which, the model is given some training examples of features that are mapped to the corresponding label. According to these examples, the model is trained to classify or predict the labels of other inputs.

- Unsupervised learning: in which, the model is trained on data and features without labels. The algorithm is used to find patterns and relationships, then it predicts the output for new input. Such kind needs more data for training and it helps to reduce the influence of human factors.

- Reinforcement learning: it does not use labeling such as in supervised and unsupervised models. Basically, it takes actions and analyses the results of these actions. It learns from linking environment's pre-condition, action and environment's post-conditions. It is popularly useful in games and robotics.

- Deep learning: it is a subset of machine learning and Artificial Intelligence (AI). it can be supervised, unsupervised, semi-supervised, self-supervised, or reinforcement-based. It uses neural networks with more than three layers to cluster similar inputs and make decisions. It is popularly useful in image recognition and natural language processing [19]. Deep learning will be used in this work.

This work uses deep learning models to handle the faults within distributed systems. In fact, a large task can be divided into many sub-tasks, which are distributed among multiple processors. A faulty processor may corrupt data, may have processing issues or may not be able to communicate properly. Therefore, deep learning model trains the model on the inputs (other sub-tasks) and the outputs of other processors to find patterns and to predict the proper output of the faulty processors. In fact, there are two kinds of input data which are structured and unstructured data. The structured data includes data sets, databases, forms, and others, while unstructured data can be images, texts, voice, and others.

This work proposes a fault tolerance technique in distributed systems using deep learning models. In which, the input is divided into sub-tasks and distributed among processors, some possessors process data successfully whereas some other do not. The faulty processors may not provide the right output or may not provide any output. Consequently, there are three scenarios as follows:

- Safe input and corrupted output.

- Corrupted input and output.

- Safe input (but unrelated) and corrupted output.

Firstly, in case of safe input and corrupted output, the sub-tasks of the faulty processors can be re-sent to other processors to get the right outputs, which is the traditional solution. Otherwise, we can use a deep learning model to learn from the inputs and the outputs of successful processors, to find patterns, and to produce the proper output for the faulty processors. Secondly, if the faulty possessors corrupt their inputs as well, then the traditional solution is not possible and the deep learning model learns from the inputs and the outputs of successful processors, to find patterns and to produce the proper output of the faulty processors, even with corrupted inputs. Thirdly, the case of safe input but unrelated and corrupted output happens when the sub-tasks (input) of the faulty processors differ from the sub-tasks of successful ones. This means the patterns of the sub-tasks of the faulty processors differ from the patterns of sub-tasks of successful ones, which challenges the learning process and the accuracy of prediction. In this case, the deep learning model is able to discover the new patterns and label them as *unknown*. The unknown label means that this data has its own label that does not exist in the learning data.

This paper uses deep learning models such as VGG16, VGG19, AlexNet LSTM and ResNet34, to investigate the performance of the deep learning models in the three mentioned scenarios. Actually, each scenario is examined using both structured and unstructured data. The used deep learning models are described below:

- VGG16: is the Visual Geometry Group which is a deep convolutional neural network model that has 16 layers [20].

- VGG19: is the Visual Geometry Group which is a deep convolutional neural network model that has 19 layers [20].

- AlexNet: is a deep convolutional neural network that has 8 layers [21].

- LSTM: Long Short Term Memory is a deep convolutional neural network model that has 3 to 4 layers. LSTM is a recurrent neural network since it has feedback connections, which enable it to learn long-term dependencies [22].

- ResNet34: Residual Neural Network is a deep convolutional neural network that has 34 layers [23].

The experimental results show that the accuracy of the mentioned deep learning models in the three scenarios is using both structured and unstructured data. For unstructured data, the results show that the accuracy of the models is affected by the size of the faulty data. The accuracy of all models lies between 60% when the size of the faulty data is 90%, and 96%, when the size of the faulty data is 90%. While the structured data is not significantly affected by the portion of the faulty data and the accuracy reaches 99% for some models. The rest of this article is organized as follows: Section 2 discusses the related work. Section 3 explains the methodology, while Section 4 shows the evaluation and experimental results. Section 6 discusses the threat of validity. Finally, Section 7 concludes the paper.

## 2 Related work

In distributed systems, the workload is distributed on multiple processors that run in parallel [24]. The advantages of using distributed system are to improve the performance and to increase the throughput. However, distributed system is challenged by many issues such as dependencies among tasks, communication cost, latency and redundancy [25]. Therefore, distributed system uses techniques such as load balancing, leader election, consensus, and fault tolerance to overcome these challenges [4, 26]. The traditional distributed system techniques do not use intelligent tools such as deep learning, where our work focuses on fault tolerance techniques using deep learning.

Researchers investigate distributed systems scalability, resilience, communication issues, and malicious attacks [27]. While others study types of distributed systems faults and failures, then they present a mapping strategy to find suitable fault tolerate techniques for each kind of faults [28, 29]. Actually, our work produces the missing and faulty output according to the correct ones.

Many works focus on fault tolerance and machine learning [30–33]. A fault tolerance framework is provided for iterative-convergent machine learning, where some miner calculation errors influence the training process. They apply fault tolerance on the calculations at some checkpoints within the training process, which reduces failure effects by 78% to 95% [30, 34]. Their techniques are suitable for numeric data, while our work uses deep learning which is more suitable for different kinds of data. Other research applies fault tolerance protocol on cloud computing using Naïve Bayes classifier to enhance reliability [35]. Another work also applies fault tolerance on cloud computing using four machine learning algorithms for job loading and failures. Support vector machine, K-nearest neighbors, logistic regression, and decision tree are used and the accuracy of all classifiers lies between 59% and 61% [36, 37]. In addition, researchers review the intelligent fault-tolerant concept that uses machine and deep learning algorithms for fault discovery and recovery [38].

However, our work uses deep learning models to achieve higher accuracy and to deal with other scenarios such as missing, corrupted, and unrelated input.

Moreover, the first model of neural network started as a one layer model in 1958 [39, 40]. After that, a multiple layers neural network model was introduced using backpropagation algorithm [41, 42]. This model enables training and classification processes. Actually, increasing the number of layers improves the accuracy of the learning model. Deep learning is one approach of a multiple layer neural network model where the number of layers is three or more. Nowadays, deep learning is widely used in the recognition and detection of images, visual objects, and speech [43, 44]. For example, researchers investigate hardware computation issues on two dimensional array computation using deep learning model. They apply fault tolerance concept to reprocess some faulty computation instead of all faults, which is enough to improve computation accuracy to the targeted threshold [45]. Moreover, deep learning results'

reliability is a very critical issue in some applications such as auto-driving, Auto-landing, and robotics [46–48]. Therefore, fault tolerant deep learning model is investigated with the consideration of different deep learning hierarchies, architectures, and layers [46]. Another work studies the performance of deep learning when it is implemented and run on a Message Passing Interface. Message Passing Interface is an interface that is designed to support parallel computing and consensus. The study strongly highlights the need for fault tolerant infrastructure to simultaneously maintain parallelism and deep learning accuracy [49]. On the other hand, our work uses deep learning models to process different kinds of data, to achieve higher accuracy, and to deal with other scenarios such as missing, corrupted, and unrelated input.

## 3 Methodology

This work applies deep learning techniques to address faults within distributed systems. As illustrated in Fig 1(a), the initial input is divided into multiple sub-tasks, which are then distributed across several processors. When a processor fails, it may either generate corrupted output or fail to produce any output at all. To mitigate this, the deep learning model leverages the inputs (other sub-tasks) and outputs from the functioning processors to identify patterns and generate the correct output, as depicted in Fig 1(b). Following the training process, as shown in Fig 1(c), the deep learning model takes the input intended for the faulty processor and predicts the correct output, effectively compensating for the fault.

Additionally, the proposed methodology evaluates the effectiveness of our model across three different scenarios. In the first scenario, where the input is correct but the output is faulty, rather than re-processing the sub-tasks of the faulty processors using other processors, we employ a deep learning model. This model learns from the inputs and outputs of the successful processors, identifies patterns, and then generates the correct output for the faulty processors, as previously illustrated in Fig 1. Secondly, when faulty processors also corrupt their inputs, the traditional approach of reprocessing sub-tasks becomes ineffective. In this scenario, our deep learning model learns from the inputs and outputs of successful processors to identify patterns and predict the correct output for the faulty processors, even when working with corrupted inputs. To simulate input corruption, we randomly remove approximately 10% of the input data, a scenario we refer to as "missing." Examples of missing data are illustrated in Figs 2 and 3. Fig 2 shows missing parts in images from a human gait dataset, while Fig 3 depicts missing sections in images representing different driver statuses (e.g., drinking, talking to a passenger, texting, etc.) in a driver distraction dataset. Additionally, we removed portions of feature values in structured data, specifically from the JavaScript vulnerability and KKD CUP99 datasets.

Thirdly, when the input is correct but the output is unrelated or corrupted, this typically occurs when the sub-tasks (input) of faulty processors differ significantly from those of the successful processors. This discrepancy means that the patterns within the faulty processors' sub-tasks are not aligned with those of the successful ones, complicating the learning process and challenging the accuracy of the deep learning model. In such cases, the deep learning model can detect these new patterns and categorize them under a new label termed as *unknown*. The unknown label indicates that the data possesses characteristics not present in the training set.

To handle this scenario, we employed an autoencoder deep learning architecture [52, 53] to identify and label the "unknown" class. Autoencoders are particularly useful for detecting anomalies or outliers. We trained the autoencoder on known, or "inlier," data to establish a baseline for expected reconstruction error. When a new observation is introduced, it is processed through the autoencoder, which computes its reconstruction error. If this error

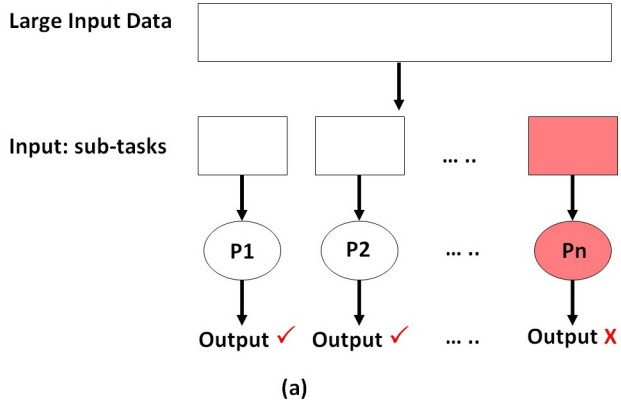

(a)

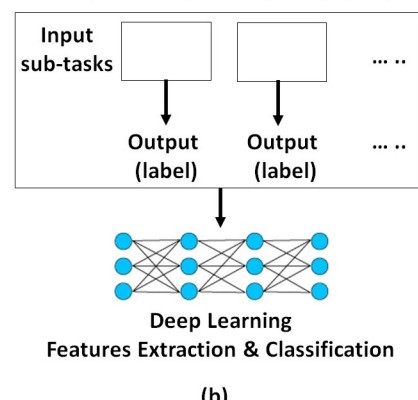

(b)

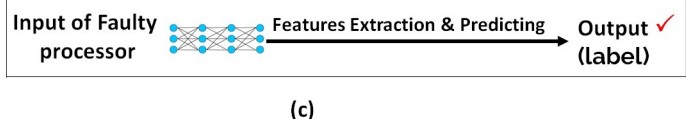

(c)

**Fig 1. Proposed Methodology: (a) Shows Phase I, where Parallel and Distributed Processing is Conducted; (b) Shows Phase II, which Includes the Training of Deep Learning Model Using None-faulty Data from Phase I, (c) Shows the Phase III, where the Output of the Faulty Processor is Produced Using Deep Learning Model.**

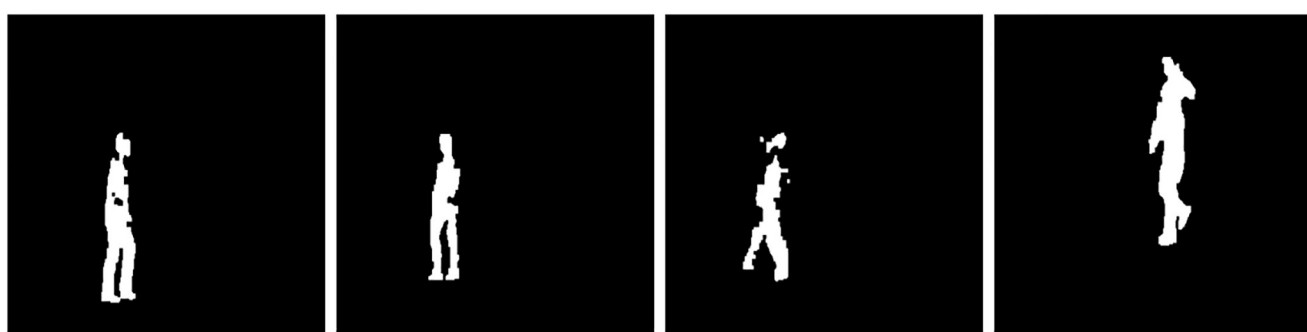

**Fig 2. Example of missing data in human gait dataset [50].**

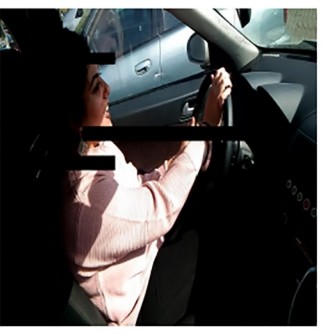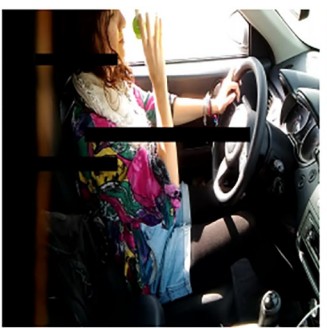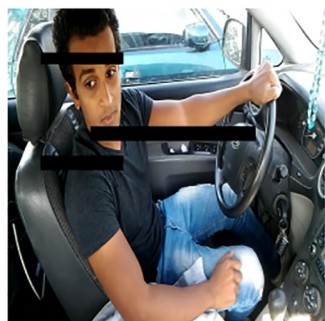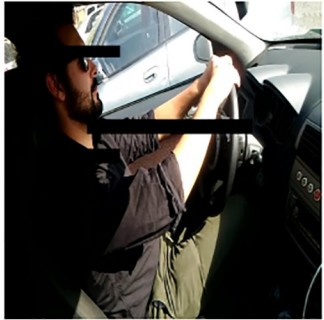

**Fig 3. Some examples of missing data in driver distraction dataset [51].**

significantly deviates from the expected range for inliers, exceeding a predefined threshold, the observation is classified as belonging to the unknown class.

This paper employs deep learning models, including VGG16, VGG19, AlexNet, and ResNet34, to evaluate their performance across the three aforementioned scenarios using image datasets. Additionally, the VGG16, VGG19, AlexNet, and LSTM models are utilized to test these scenarios on two textual datasets. Indeed, LSTM is chosen for structured data due to its sequential nature, which makes it particularly well-suited for handling structured features.

## 4 Evaluation

### 4.1 Dataset description

There are two kinds of input data which are structured and unstructured data. The structured data includes data sets, databases, forms, and others, while unstructured data can be images, texts, voice, and others. In this work, we evaluate our approach using four publicly available datasets, where two datasets are based on images which are distracted driver and Gait human datasets (unstructured datasets), and the other two datasets are based on traditional features such as JavaScript Vulnerability and Intrusion detection system (KDD Cup 99) datasets (structured datasets). The details of the datasets are given in Table 1.

The first dataset we used in this experiment is the distracted driver dataset [51, 54], which contains 17,309 frames distributed over the following classes: Safe Driving (3,686), Phone Right (1,223), Phone Left (1,361), Text Right (1,974), Text Left (1,301), Adjusting Radio (1,220), Drinking (1,612), Hair or Makeup (1,202), Reaching Behind (1,159), and Talking to Passenger (2,570).

The second dataset we used is Gait Recognition (CASIA-A). CASIA-A has 19,135 silhouettes with different walking positions. It consists of twenty classes. Each class in a different name and different images and has different walking corners Created by Wang et al. [50].

**Table 1. Datasets description.**

| Reference | Dataset Name | Size | Attributes | Classes | Data Type |
|---|---|---|---|---|---|
| [51, 54] | Distracted Driver | 17,309 | Feature discriptors | 10 | Images |
| [50] | Gait Recognition (CASIA-A) | 19,135 | Feature discriptors | 20 | Images |
| [55] | JavaScript vulnerability | 12,125 | 35 | 2 | Numbers |
| [56] | KDD CUP 99 | 67,343 | 39 | 5 | Numbers |

The third dataset we used is JavaScript vulnerability dataset (Ferenc et al's dataset [55]) from the Node Security Project. Ferenc et al's dataset has 12,125 functions. It consists of two classes. Vulnerability functions contain 1,496 and non-vulnerability functions contain 10,629 functions.

The fourth dataset used in this experiment is the KDD CUP 99 dataset [56], which contains normal data 67,343 samples and four types of attacks, namely, the denial of service (DOS) contains 45927 samples attacks, remote-to-local (R2L) contains 995 samples attacks, user-to-privilege (U2R) contain 52 samples, and needle attack (probe) contains 11,656 samples attacks. Each piece of data contains 41 features [57]. The function description of each feature [58–60].

## 4.2 Performance measurements

On the other hand, our deep learning models are used to process the mentioned datasets to evaluate the results of our assumptions in the three scenarios. Classification accuracy is commonly used to assess the performance of deep learning models. Actually, a variety of performance measurements are shown to see different sides of the results and to get deep insight. The main measurement is the *accuracy* of the deep learning. The *precision*, *recall*, $F_1 - score$, and *AUC* are also presented to evaluate our results. In addition, *lossfunction* is also used with the structured datasets. We also compared prediction accuracy and failures using confusion matrices. The preceding equations' terms *TP*, *TN*, *FP*, and *FN* signify true positives, true negatives, false positives, and false negatives, respectively.

The accuracy and F1-score are the primary performance measure for all of the classifier models used in our research. The fraction of accurate predictions to all input samples is measured by a single statistic known as the F1-score. The accuracy or actual positive rate (TPR) is the number of accurate positive outcomes divided by the number of positive results predicted by the classifier. This can be defined as follows: Precision is defined as $TP/(TP + FP)$, where TP is the number of true positives or the correct forecast of a positive sample, and FP is the number of false positives. The results of all equations are between 0 and 1.

In formula 5, *H* is referred to cross entropy loss function, where $p(x)$ is the true distribution, and $q(x)$, is the estimated distribution, defined over the discrete variable *x* and is given by formula 5.

$$Accuracy = \frac{TP + TN}{TP + FP + TN + FN} \tag{1}$$

$$Precision(P) = \frac{TP}{TP + FP} \tag{2}$$

$$Recall(R) = \frac{TP}{TP + FN} \tag{3}$$

$$F_1 - score = \frac{2 * Precision * Recall}{Precision + Recall} \tag{4}$$

$$H(p, q) = -\sum_{\forall x} p(x) \times log(q(x)) \tag{5}$$

## 5 Experimental results

This study aims to investigate the impact of datasets types and size on the classification performance and recommend the appropriate models for limited-size datasets to handle the faults within distributed systems. The experiments examine the scenarios of faults as follows:

1. In case of safe input and corrupted output. The dataset is divided into two groups, one represents the none-faulty data which will be the training set, and the other is the faulty data which is used as a testing set. We examine different sizes of faulty data starting from 10% to 90% of the whole data.

2. In case of corrupted input and corrupted output, part of the inputs (testing set) is missing as explained earlier. We examine different sizes of faulty data starting from 10% to 90% of the whole data. Our goal is to build a deep learning model that can identify corrupted images. Also, we are able to clean the dataset and apply artifact cleaning. We create random distortion on corrected images to generate a dataset big enough for our experiments. We then build deep learning models based on Convolutional Neural Networks (CNNs) to identify and classify the corrupted images.

3. In case of unrelated input and corrupted output, where the unrelated input is classified as *unknown*. This indicates that the input does not fit under the classes of the training phase. In fact, we conduct the training and then run the classification for the testing set (faulty data). After that, we detect the outliers and classify them as unrelated (*unknown*), instead of enforcing them under unsuitable class.

4. The experiments examine one more scenario that combines both scenarios in points two and three, where the input is corrupted and unrelated at the same time.

For each mentioned scenario we test different sizes of faults, where the percentages of the faulty data can be 10%, 20%, 30%, 40%, 50%, 60%, 70%, 80% or 90% of the whole data. We examine the impact of reducing the size of the training set on the classification performance. After pre-processing the datasets, deep learning models were trained on all datasets. The performance of the classifiers is evaluated with respect to accuracy, precision, recall, specificity, f-score, and AUC.

Four deep learning models, namely, VGG16, VGG19, AlexNet, and ResNet34 are used, to study the performance of the deep learning models in the mentioned scenarios and on different fault sizes using two different images datasets. In addition, VGG16, VGG19, AlexNet, and LSTM deep learning models are used to test the first two mentioned scenarios with different fault sizes on the other two textual datasets.

### 5.1 Unstructured datasets

In this part, we run the four mentioned scenarios using the four deep learning models, namely, VGG16, VGG19, AlexNet and ResNet34, with different sizes of faulty results, using two unstructured datasets. For the first dataset, namely, the distracted driver dataset, the accuracy is measured as illustrated in Fig 4. The figure shows that for VGG16, the accuracy of the first scenario (when the input is safe) reaches 90%, when the size of faulty data (testing set) is 10% and the accurate data (that is used as a training sets) is 90%. By increasing the size of the faulty data and decreasing the size of the training set, the accuracy is gradually dropped until it becomes 69%, when the faulty data is 90%. The second scenario deals with corrupted input by having some missing data scenario. The VGG16 accuracy reaches 92%, when the size of faulty data (testing set) is 10% and the accurate data (that is used as a training sets) is 90%. By increasing the size of the faulty data and decreasing the size of the training set, the accuracy is

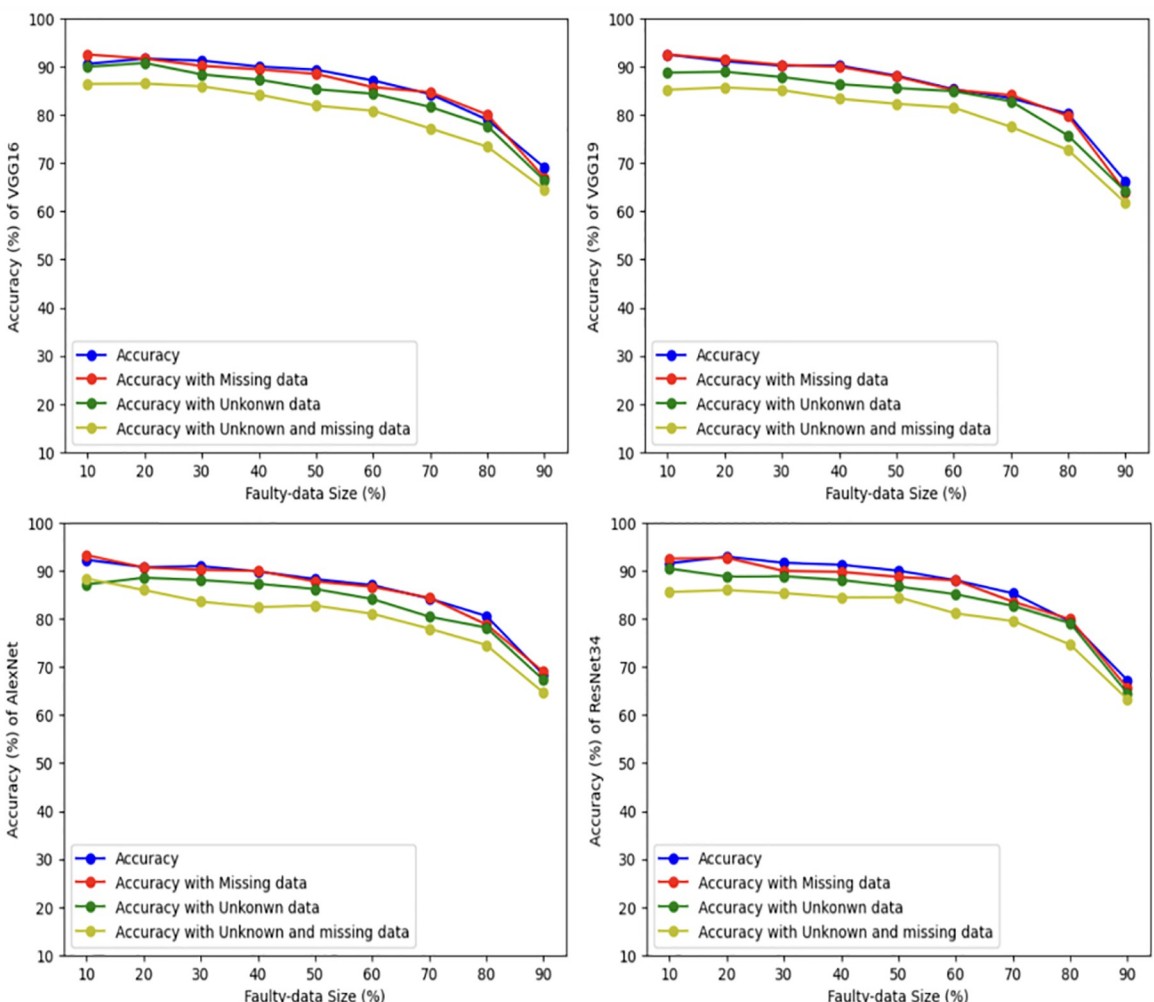

**Fig 4. Driver distraction dataset results using four deep learning models.**

gradually dropped until it reaches 67%, when the faulty data is 90%. The third scenario has unrelated input that is classified as *unknown*. In which, the VGG16 accuracy reaches 89%, when the faulty data is 10%, and then decreases to 66%, when the size of faulty data is 90%. The fourth scenario uses corrupted (missing) and unrelated input, where the VGG16 accuracy reaches 84%, when the faulty data is 10%, and then decreases to 64%, when the size of faulty data is 90%.

The same thing is applied to VGG19, AlexNet, and ResNet34. In general, the accuracy of the first scenario is the best in most tests since the input (testing set) is safe. The second scenario reaches almost the same accuracy in comparison to the first one even with some missing input of the testing set. The accuracy of the deep learning models for the third and fourth scenarios is a little less in comparison to the first scenario, however all scenarios are very close and still acceptable.

Table 2 gives more insight views for the performance of the four deep learning models by presenting more supportive details about precision, recall, specificity, f-score, and AUC.

**Table 2. Driver distraction results.**

| Model | Precision | Precision* | Precision** | Prceision*** | Recall | Recall* | Recall** | Recall*** | F1-score | F1-score* | F1-score** | F1-score*** | AUC | AUC* | AUC** | AUC*** |
|---|---|---|---|---|---|---|---|---|---|---|---|---|---|---|---|---|
| **Driver Distraction Models with 10% of Training size** | | | | | | | | | | | | | | | | |
| VGG16 | 69.38 | 67.45 | 65.82 | 64.29 | 69.09 | 67.01 | 66.42 | 64.56 | 69.13 | 66.68 | 65.73 | 64.19 | 69.09 | 67.01 | 66.42 | 64.56 |
| VGG19 | 66.22 | 64.81 | 64.27 | 62.43 | 66.24 | 64.10 | 64.15 | 61.77 | 66.12 | 63.83 | 64.11 | 61.91 | 66.24 | 64.10 | 64.15 | 61.77 |
| AlexNet | 69.77 | 69.30 | 66.95 | 64.13 | 68.43 | 69.00 | 67.33 | 64.67 | 68.31 | 68.92 | 66.74 | 63.89 | 68.43 | 69.00 | 67.33 | 64.67 |
| ResNet34 | 68.47 | 66.19 | 66.19 | 64.19 | 63.26 | 67.26 | 65.69 | 64.57 | 63.30 | 67.2165.61 | 63.92 | 62.92 | 67.26 | 65.69 | 64.57 | 63.30 |
| **Driver Distraction Models with 20% of Training Size** | | | | | | | | | | | | | | | | |
| VGG16 | 79.05 | 79.97 | 76.91 | 72.79 | 79.02 | 80.06 | 77.71 | 73.41 | 78.91 | 79.91 | 77.05 | 72.94 | 79.02 | 80.06 | 77.71 | 73.41 |
| VGG19 | 80.31 | 79.93 | 75.27 | 72.43 | 80.24 | 79.77 | 75.72 | 72.73 | 80.19 | 79.75 | 75.38 | 72.48 | 80.24 | 79.77 | 75.72 | 72.73 |
| AlexNet | 80.84 | 79.10 | 77.42 | 74.33 | 80.62 | 78.85 | 78.20 | 74.58 | 80.62 | 78.81 | 77.58 | 74.30 | 80.62 | 78.85 | 78.20 | 74.58 |
| ResNet34 | 80.01 | 80.17 | 78.66 | 74.94 | 79.49 | 79.95 | 79.11 | 74.67 | 79.51 | 79.90 | 78.72 | 74.66 | 79.49 | 79.95 | 79.11 | 74.67 |
| **Driver Distraction Models with 30% of Training Size** | | | | | | | | | | | | | | | | |
| VGG16 | 84.24 | 84.86 | 81.20 | 77.11 | 84.32 | 84.73 | 81.67 | 77.21 | 84.24 | 84.70 | 81.36 | 76.97 | 84.32 | 84.73 | 81.67 | 77.21 |
| VGG19 | 83.50 | 84.25 | 82.11 | 77.44 | 83.50 | 84.15 | 82.84 | 77.54 | 83.42 | 84.09 | 82.27 | 77.39 | 83.50 | 84.15 | 82.84 | 77.54 |
| AlexNet | 84.65 | 84.65 | 79.84 | 77.53 | 84.27 | 84.43 | 80.49 | 77.99 | 84.29 | 84.45 | 79.96 | 77.54 | 84.27 | 84.43 | 80.49 | 77.99 |
| ResNet34 | 85.56 | 83.84 | 82.26 | 79.18 | 85.35 | 83.59 | 82.74 | 79.58 | 85.36 | 83.60 | 82.36 | 79.19 | 85.35 | 83.59 | 82.74 | 79.58 |
| **Driver Distraction Models with 40% of Training Size** | | | | | | | | | | | | | | | | |
| VGG16 | 85.96 | 85.96 | 83.86 | 80.80 | 85.76 | 85.76 | 84.40 | 80.88 | 85.80 | 85.80 | 84.05 | 80.72 | 85.76 | 85.76 | 84.40 | 80.88 |
| VGG19 | 85.53 | 85.23 | 84.30 | 81.69 | 85.35 | 85.19 | 84.938 | 81.53 | 85.35 | 85.17 | 84.54 | 81.50 | 85.35 | 85.19 | 84.93 | 81.53 |
| AlexNet | 87.48 | 86.95 | 83.81 | 80.84 | 87.06 | 86.72 | 84.17 | 81.10 | 87.00 | 86.72 | 83.88 | 80.67 | 87.06 | 86.72 | 84.17 | 81.10 |
| ResNet34 | 88.36 | 88.21 | 84.80 | 80.76 | 88.06 | 88.04 | 85.15 | 81.16 | 88.10 | 88.02 | 84.79 | 80.86 | 88.06 | 88.04 | 85.15 | 81.16 |
| **Driver Distraction Models with 50% of Training Size** | | | | | | | | | | | | | | | | |
| VGG16 | 88.62 | 88.62 | 85.04 | 82.31 | 88.53 | 88.53 | 85.35 | 81.93 | 88.51 | 88.51 | 85.12 | 81.87 | 88.53 | 88.53 | 85.35 | 81.93 |
| VGG19 | 88.16 | 88.07 | 85.29 | 82.70 | 88.16 | 87.99 | 85.6 | 82.32 | 88.14 | 87.97 | 85.39 | 82.28 | 88.16 | 87.99 | 85.60 | 82.32 |
| AlexNet | 88.36 | 87.99 | 85.95 | 82.59 | 88.33 | 87.82 | 86.25 | 82.81 | 88.31 | 87.78 | 85.87 | 82.40 | 88.33 | 87.82 | 86.25 | 82.81 |
| ResNet34 | 90.14 | 88.85 | 86.31 | 84.23 | 90.06 | 88.75 | 86.77 | 84.52 | 90.06 | 88.73 | 86.46 | 84.31 | 90.06 | 88.75 | 86.77 | 84.52 |
| **Driver Distraction Models with 60% of Training Size** | | | | | | | | | | | | | | | | |
| VGG16 | 89.58 | 89.58 | 86.95 | 84.56 | 89.47 | 89.47 | 87.34 | 84.22 | 89.46 | 89.46 | 87.08 | 84.06 | 89.47 | 89.47 | 87.34 | 84.22 |
| VGG19 | 90.27 | 90.17 | 86.47 | 83.15 | 90.27 | 90.02 | 86.94 | 83.37 | 90.24 | 90.03 | 86.57 | 83.05 | 90.27 | 90.02 | 86.94 | 83.37 |
| AlexNet | 90.08 | 90.04 | 86.66 | 83.24 | 89.90 | 90.02 | 87.34 | 82.48 | 89.90 | 90.01 | 86.91 | 82.20 | 89.90 | 90.02 | 87.34 | 82.48 |
| ResNet34 | 91.39 | 89.91 | 87.77 | 83.66 | 91.30 | 89.81 | 88.10 | 83.86 | 91.30 | 89.81 | 87.83 | 83.71 | 91.30 | 89.81 | 88.10 | 83.86 |
| **Driver Distraction Models with 70% of Training Size** | | | | | | | | | | | | | | | | |
| VGG16 | 90.26 | 90.26 | 88.04 | 84.68 | 90.20 | 90.20 | 88.41 | 84.95 | 90.18 | 90.18 | 88.18 | 84.63 | 90.20 | 90.20 | 88.41 | 84.95 |
| VGG19 | 90.32 | 90.44 | 87.64 | 85.05 | 90.24 | 90.40 | 87.88 | 85.15 | 90.22 | 90.38 | 87.65 | 84.89 | 90.24 | 90.40 | 87.88 | 85.15 |
| AlexNet | 91.10 | 90.36 | 87.58 | 83.35 | 91.01 | 90.24 | 88.12 | 83.60 | 90.99 | 90.23 | 87.80 | 83.31 | 91.01 | 90.24 | 88.12 | 83.60 |

*(Continued)*

**Table 2.** (Continued)

| Model | Precision | Precision* | Precision** | Prceision*** | Recall | Recall* | Recall** | Recall*** | F1-score | F1-score* | F1-score** | F1-score*** | AUC | AUC* | AUC** | AUC*** |
|---|---|---|---|---|---|---|---|---|---|---|---|---|---|---|---|---|
| ResNet34 | 91.71 | 90.11 | 88.47 | 84.87 | 91.70 | 89.99 | 88.89 | 85.39 | 91.69 | 89.99 | 88.62 | 85.04 | 91.70 | 89.99 | 88.89 | 85.39 |
| **Driver Distraction Models with 80% of Training Size** | | | | | | | | | | | | | | | | |
| VGG16 | 91.73 | 91.73 | 90.63 | 86.08 | 91.70 | 91.70 | 90.79 | 86.52 | 91.70 | 91.70 | 90.68 | 86.14 | 91.70 | 91.70 | 90.79 | 86.52 |
| VGG19 | 91.21 | 91.60 | 88.47 | 85.91 | 91.15 | 91.52 | 88.96 | 85.72 | 91.16 | 91.51 | 88.57 | 85.41 | 91.15 | 91.52 | 88.96 | 85.72 |
| AlexNet | 90.91 | 90.86 | 88.16 | 85.93 | 90.79 | 90.73 | 88.59 | 86.03 | 90.77 | 90.71 | 88.164 | 85.77 | 90.79 | 90.73 | 88.59 | 86.03 |
| ResNet34 | 93.16 | 92.83 | 88.68 | 85.57 | 92.98 | 92.74 | 88.83 | 86.03 | 93.00 | 92.76 | 88.68 | 85.65 | 92.98 | 92.74 | 88.83 | 86.03 |
| Model | Precision | Precision* | Precision** | Prceision*** | Recall | Recall* | Recall** | Recall*** | F1-score | F1-score* | F1-score** | F1-score*** | AUC | AUC* | AUC** | AUC*** |
| **Driver Distraction Models with 90% of Training Size** | | | | | | | | | | | | | | | | |
| VGG16 | 92.60 | 92.60 | 90.63 | 86.70 | 92.56 | 92.56 | 90.79 | 86.46 | 92.53 | 92.53 | 90.68 | 86.51 | 92.56 | 92.56 | 90.79 | 86.46 |
| VGG19 | 92.58 | 92.65 | 88.74 | 85.87 | 92.56 | 92.56 | 88.78 | 85.24 | 92.54 | 92.55 | 88.65 | 85.07 | 92.56 | 92.56 | 88.78 | 85.24 |
| AlexNet | 92.46 | 93.48 | 87.14 | 88.88 | 92.32 | 93.29 | 87.20 | 88.41 | 92.33 | 93.33 | 87.09 | 88.44 | 92.31 | 93.29 | 87.20 | 88.41 |
| ResNet34 | 91.90 | 92.76 | 86.50 | 90.03 | 91.59 | 92.56 | 85.61 | 90.49 | 91.57 | 92.52 | 85.86 | 90.17 | 91.59 | 92.56 | 85.61 | 90.49 |

* Missing data results.

** Unknown data results.

*** Missing and unknown data results.

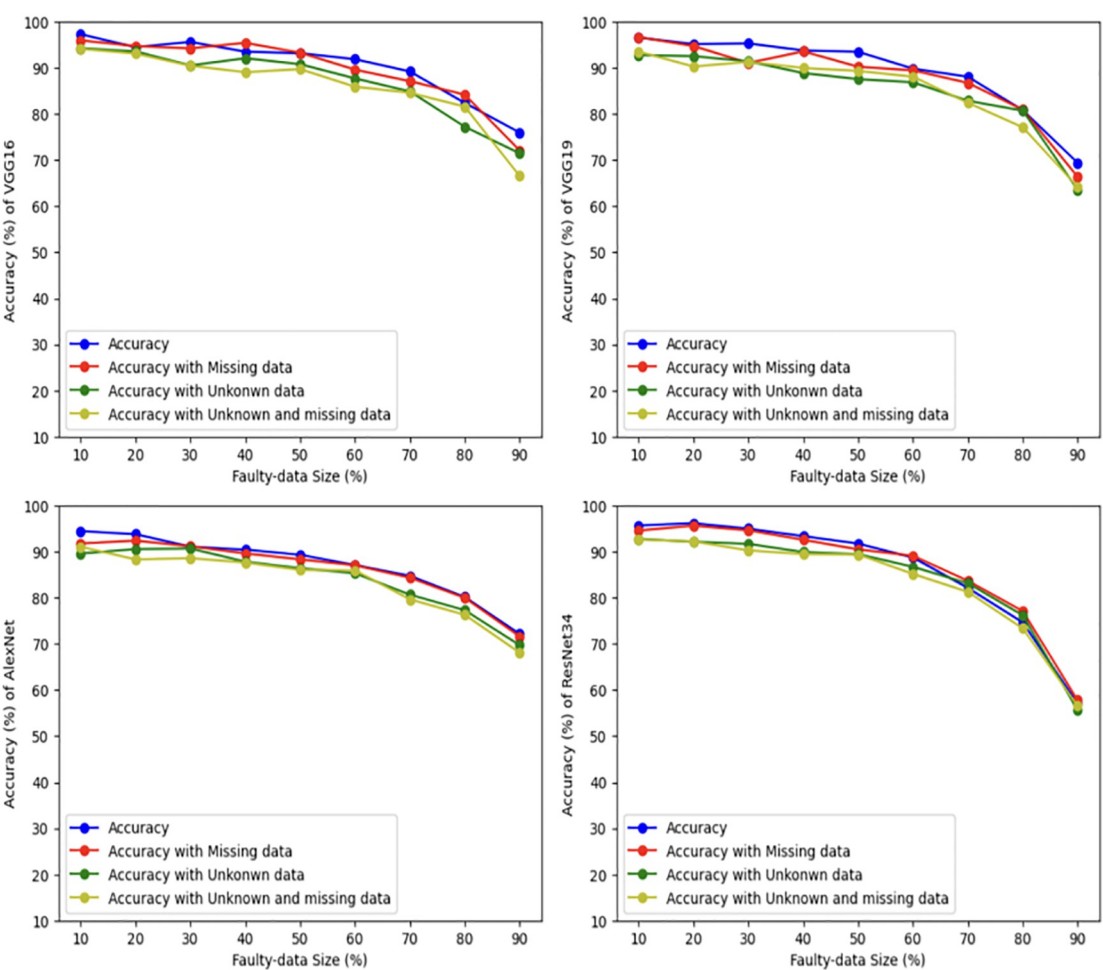

**Fig 5. Gait human dataset results using four deep learning models.**

For the second dataset, namely, the Gait Recognition dataset, the accuracy is shown in Fig 5. The figure shows that for VGG16 the accuracy of the first scenario reaches 97%, when the size of the faulty data is 10% and the testing set is 90%. Then it decreases to 77%, when the faulty data is 90%. Second scenario accuracy reaches 95%, when the size of the faulty data is 10%. It drops to 71%, when the size of the faulty data 90%. In the third scenario, the VGG16 accuracy reaches 95%, when the faulty data is 10%, and then decreases to 71%, when the size of faulty data is 90%. In the fourth scenario, the VGG16 accuracy reaches 94%, when the faulty data is 10%, and then decreases to 66%, when the size of faulty data is 90%.

For VGG19, the accuracy of all scenarios lay between 95% and 92%, when the size of faulty data is 10%. Then they decrease until the accuracy becomes between 72% and 68%, by increasing the faulty data size to 90%. Almost similar results are applied for AlexNet. However, ResNet34 has less accuracy, as it drops to about 56% when the faulty data size becomes 90%.

Table 3 gives more insight views for the performance of the four deep learning models by presenting more supportive details about precision, recall, specificity, f-score, and AUC.

**Table 3. Human gait detection results.**

**Human Gait Models with 10% of Training size**

| Model | Precision | Precision* | Precision** | Prceision*** | Recall | Recall* | Recall** | Recall*** | F1-score | F1-score* | F1-score** | F1-score*** | AUC | AUC* | AUC** | AUC*** |
|---|---|---|---|---|---|---|---|---|---|---|---|---|---|---|---|---|
| VGG16 | 76.20 | 73.43 | 72.83 | 67.54 | 75.99 | 72.17 | 71.48 | 66.58 | 75.80 | 71.92 | 71.02 | 66.25 | 75.99 | 72.17 | 71.48 | 66.58 |
| VGG19 | 70.20 | 68.76 | 65.75 | 65.46 | 69.43 | 66.40 | 63.61 | 64.22 | 69.32 | 66.31 | 63.59 | 63.72 | 69.43 | 66.40 | 63.61 | 64.22 |
| AlexNet | 72.96 | 72.74 | 70.20 | 69.33 | 72.20 | 71.65 | 69.75 | 68.11 | 72.06 | 71.49 | 69.53 | 68.14 | 72.20 | 71.65 | 69.75 | 68.11 |
| ResNet34 | 57.64 | 58.34 | 56.30 | 57.62 | 57.10 | 57.81 | 55.62 | 56.61 | 56.99 | 57.78 | 55.74 | 56.83 | 57.10 | 57.81 | 55.62 | 56.61 |

**Human Gait Models with 20% of Training Size**

| Model | Precision | Precision* | Precision** | Prceision*** | Recall | Recall* | Recall** | Recall*** | F1-score | F1-score* | F1-score** | F1-score*** | AUC | AUC* | AUC** | AUC*** |
|---|---|---|---|---|---|---|---|---|---|---|---|---|---|---|---|---|
| VGG16 | 83.48 | 84.85 | 78.11 | 81.61 | 82.42 | 84.21 | 77.25 | 81.61 | 82.42 | 84.19 | 77.06 | 81.42 | 82.42 | 84.21 | 77.25 | 81.61 |
| VGG19 | 82.19 | 81.80 | 81.01 | 78.14 | 80.85 | 80.96 | 80.73 | 77.13 | 80.98 | 80.99 | 80.49 | 77.15 | 80.85 | 80.96 | 80.73 | 77.13 |
| AlexNet | 81.59 | 80.52 | 77.75 | 76.96 | 80.21 | 80.03 | 77.34 | 76.34 | 80.16 | 79.92 | 77.04 | 75.61 | 80.21 | 80.03 | 77.34 | 76.34 |
| ResNet34 | 74.83 | 77.27 | 76.73 | 73.62 | 74.67 | 77.14 | 76.23 | 73.38 | 74.64 | 77.00 | 76.06 | 73.33 | 74.67 | 77.14 | 76.23 | 73.38 |

**Human Gait Models with 30% of Training Size**

| Model | Precision | Precision* | Precision** | Prceision*** | Recall | Recall* | Recall** | Recall*** | F1-score | F1-score* | F1-score** | F1-score*** | AUC | AUC* | AUC** | AUC*** |
|---|---|---|---|---|---|---|---|---|---|---|---|---|---|---|---|---|
| VGG16 | 89.44 | 87.94 | 84.89 | 84.88 | 89.24 | 87.13 | 84.92 | 84.62 | 89.28 | 87.18 | 84.62 | 84.55 | 89.24 | 87.13 | 84.92 | 84.62 |
| VGG19 | 88.69 | 87.36 | 82.94 | 82.71 | 88.09 | 86.75 | 82.88 | 82.44 | 88.06 | 86.76 | 82.51 | 82.15 | 88.09 | 86.75 | 82.88 | 82.44 |
| AlexNet | 85.59 | 85.05 | 81.33 | 81.33 | 84.80 | 84.38 | 80.71 | 79.69 | 84.95 | 84.48 | 80.26 | 79.67 | 84.80 | 84.38 | 80.71 | 79.69 |
| ResNet34 | 82.20 | 83.76 | 83.25 | 81.44 | 82.16 | 83.71 | 83.16 | 81.31 | 82.08 | 83.69 | 83.11 | 81.29 | 82.16 | 83.71 | 83.16 | 81.31 |

**Human Gait Models with 40% of Training Size**

| Model | Precision | Precision* | Precision** | Prceision*** | Recall | Recall* | Recall** | Recall*** | F1-score | F1-score* | F1-score** | F1-score*** | AUC | AUC* | AUC** | AUC*** |
|---|---|---|---|---|---|---|---|---|---|---|---|---|---|---|---|---|
| VGG16 | 92.05 | 90.28 | 87.59 | 86.70 | 91.90 | 89.63 | 87.73 | 85.92 | 91.92 | 89.62 | 87.50 | 85.79 | 91.90 | 89.63 | 87.73 | 85.92 |
| VGG19 | 90.10 | 90.19 | 86.97 | 87.62 | 89.85 | 89.48 | 86.88 | 88.10 | 89.85 | 89.57 | 86.75 | 87.58 | 89.85 | 89.48 | 86.88 | 88.10 |
| AlexNet | 87.76 | 87.39 | 85.50 | 85.84 | 87.14 | 87.08 | 85.35 | 85.98 | 87.26 | 87.11 | 85.20 | 85.73 | 87.14 | 87.08 | 85.35 | 85.98 |
| ResNet34 | 88.75 | 89.24 | 86.84 | 85.14 | 88.75 | 89.14 | 86.74 | 85.19 | 88.68 | 89.14 | 86.71 | 85.01 | 88.75 | 89.14 | 86.74 | 85.19 |

**Human Gait Models with 50% of Training Size**

| Model | Precision | Precision* | Precision** | Prceision*** | Recall | Recall* | Recall** | Recall*** | F1-score | F1-score* | F1-score** | F1-score*** | AUC | AUC* | AUC** | AUC*** |
|---|---|---|---|---|---|---|---|---|---|---|---|---|---|---|---|---|
| VGG16 | 93.47 | 93.51 | 90.49 | 89.19 | 93.21 | 93.31 | 90.82 | 89.75 | 93.21 | 93.32 | 90.32 | 89.33 | 93.21 | 93.31 | 90.82 | 89.75 |
| VGG19 | 93.68 | 90.56 | 87.59 | 89.29 | 93.51 | 90.26 | 87.57 | 89.36 | 93.52 | 90.29 | 87.48 | 89.19 | 93.51 | 90.26 | 87.57 | 89.36 |
| AlexNet | 89.57 | 88.74 | 86.19 | 86.17 | 89.36 | 88.35 | 86.45 | 86.08 | 89.37 | 88.35 | 86.14 | 85.93 | 89.36 | 88.35 | 86.45 | 86.08 |
| ResNet34 | 91.97 | 90.60 | 89.18 | 89.50 | 91.80 | 90.53 | 89.48 | 89.43 | 91.83 | 90.54 | 89.26 | 89.41 | 91.80 | 90.53 | 89.48 | 89.43 |

**Human Gait Models with 60% of Training Size**

| Model | Precision | Precision* | Precision** | Prceision*** | Recall | Recall* | Recall** | Recall*** | F1-score | F1-score* | F1-score** | F1-score*** | AUC | AUC* | AUC** | AUC*** |
|---|---|---|---|---|---|---|---|---|---|---|---|---|---|---|---|---|
| VGG16 | 93.70 | 95.48 | 91.91 | 89.11 | 93.53 | 95.45 | 92.10 | 89.08 | 93.50 | 95.45 | 91.92 | 88.87 | 93.53 | 95.45 | 92.10 | 89.08 |
| VGG19 | 94.03 | 93.75 | 89.28 | 90.07 | 93.81 | 93.59 | 88.89 | 89.99 | 93.84 | 93.61 | 88.82 | 89.81 | 93.81 | 93.59 | 88.89 | 89.99 |
| AlexNet | 90.83 | 90.03 | 88.21 | 87.45 | 90.48 | 89.66 | 87.85 | 87.67 | 90.55 | 89.71 | 87.92 | 87.40 | 90.48 | 89.66 | 87.85 | 87.67 |
| ResNet34 | 93.45 | 92.63 | 89.88 | 89.49 | 93.41 | 92.62 | 89.96 | 89.50 | 93.39 | 92.59 | 89.89 | 89.43 | 93.41 | 92.62 | 89.96 | 89.50 |

**Human Gait Models with 70% of Training Size**

| Model | Precision | Precision* | Precision** | Prceision*** | Recall | Recall* | Recall** | Recall*** | F1-score | F1-score* | F1-score** | F1-score*** | AUC | AUC* | AUC** | AUC*** |
|---|---|---|---|---|---|---|---|---|---|---|---|---|---|---|---|---|
| VGG16 | 95.69 | 94.43 | 90.69 | 90.45 | 95.65 | 94.26 | 90.56 | 90.48 | 95.64 | 94.28 | 90.37 | 90.22 | 95.65 | 94.26 | 90.56 | 90.48 |
| VGG19 | 95.41 | 91.50 | 91.01 | 91.50 | 95.32 | 91.09 | 91.42 | 91.33 | 95.33 | 91.11 | 91.05 | 91.29 | 95.32 | 91.09 | 91.42 | 91.33 |
| AlexNet | 91.50 | 91.60 | 90.66 | 88.46 | 91.13 | 91.21 | 90.72 | 88.61 | 91.17 | 91.26 | 90.42 | 88.34 | 91.13 | 91.21 | 90.72 | 88.61 |

(*Continued*)

**Table 3.** (Continued)

| ResNet34 | 95.11 | 94.73 | 91.69 | 90.55 | 95.04 | 94.67 | 91.74 | 90.32 | 95.03 | 94.67 | 91.68 | 90.39 | 95.04 | 94.67 | 91.74 | 90.32 |
|---|---|---|---|---|---|---|---|---|---|---|---|---|---|---|---|---|

| Model | Precision | Precision* | Precision** | Prceision*** | Recall | Recall* | Recall** | Recall*** | F1-score | F1-score* | F1-score** | F1-score*** | AUC | AUC* | AUC** | AUC*** |
|---|---|---|---|---|---|---|---|---|---|---|---|---|---|---|---|---|
| **Human Gait Models with 80% of Training Size** | | | | | | | | | | | | | | | | |
| VGG16 | 94.90 | 95.00 | 93.47 | 92.92 | 94.51 | 94.75 | 93.59 | 93.11 | 94.53 | 94.81 | 93.49 | 92.92 | 94.51 | 94.75 | 93.59 | 93.11 |
| VGG19 | 95.28 | 94.93 | 92.43 | 89.94 | 95.18 | 94.75 | 92.56 | 90.30 | 95.18 | 94.78 | 92.31 | 89.99 | 95.18 | 94.75 | 92.56 | 90.30 |
| AlexNet | 93.98 | 92.96 | 90.45 | 87.47 | 93.84 | 92.43 | 90.60 | 88.35 | 93.85 | 92.51 | 90.33 | 87.38 | 93.84 | 92.43 | 90.60 | 88.35 |
| ResNet34 | 96.26 | 95.73 | 92.53 | 92.18 | 96.22 | 95.67 | 92.19 | 92.25 | 96.22 | 95.66 | 92.24 | 92.15 | 96.22 | 95.67 | 92.19 | 92.25 |

| Model | Precision | Precision* | Precision** | Prceision*** | Recall | Recall* | Recall** | Recall*** | F1-score | F1-score* | F1-score** | F1-score*** | AUC | AUC* | AUC** | AUC*** |
|---|---|---|---|---|---|---|---|---|---|---|---|---|---|---|---|---|
| **Human Gait Models with 90% of Training Size** | | | | | | | | | | | | | | | | |
| VGG16 | 97.35 | 96.17 | 94.11 | 93.91 | 97.32 | 95.98 | 94.27 | 94.15 | 97.32 | 95.95 | 94.16 | 93.88 | 97.32 | 95.98 | 94.27 | 94.15 |
| VGG19 | 96.64 | 96.82 | 92.71 | 93.50 | 96.59 | 96.71 | 92.80 | 93.54 | 96.59 | 96.70 | 92.59 | 93.35 | 96.59 | 96.71 | 92.80 | 93.54 |
| AlexNet | 94.72 | 92.38 | 89.94 | 90.97 | 94.51 | 91.83 | 89.63 | 91.10 | 94.54 | 91.82 | 89.40 | 90.82 | 94.51 | 91.83 | 89.63 | 91.10 |
| ResNet34 | 95.77 | 94.75 | 92.95 | 92.47 | 95.73 | 94.63 | 92.80 | 92.68 | 95.71 | 94.58 | 92.83 | 92.54 | 95.73 | 94.63 | 92.80 | 92.68 |

* Missing data results.

** Unknown data results.

*** Missing and unknown data results.

## 5.2 Structured datasets

For more verification, the vulnerability dataset is used to conduct the test of the same scenarios using VGG16, VGG19, AlexNet, and LSTM deep learning models, with different portions of fault size. Fig 6 demonstrates vulnerability dataset accuracy results using the four deep learning models. Actually, in this part, we focus only on the first two scenarios. The accuracy of both scenarios is almost the same using all models. Indeed, VGG16, VGG19, AlexNet outperform LSTM. In addition, Fig 7 shows the best Vulnerability dataset results based on loss function. In which, a loss function measures how good the deep learning model does in terms of being able to match the expected output. Therefore, the loss function is used with the structured datasets since the expected output is present. The experiment shows that the loss function decreases as the faulty data size decreases and the number of epochs increases with all models. Moreover, Fig 8 illustrates the best missing vulnerability dataset results based on the loss function, where the loss function decreases as the faulty data size decreases and the number of epochs increases with all models.

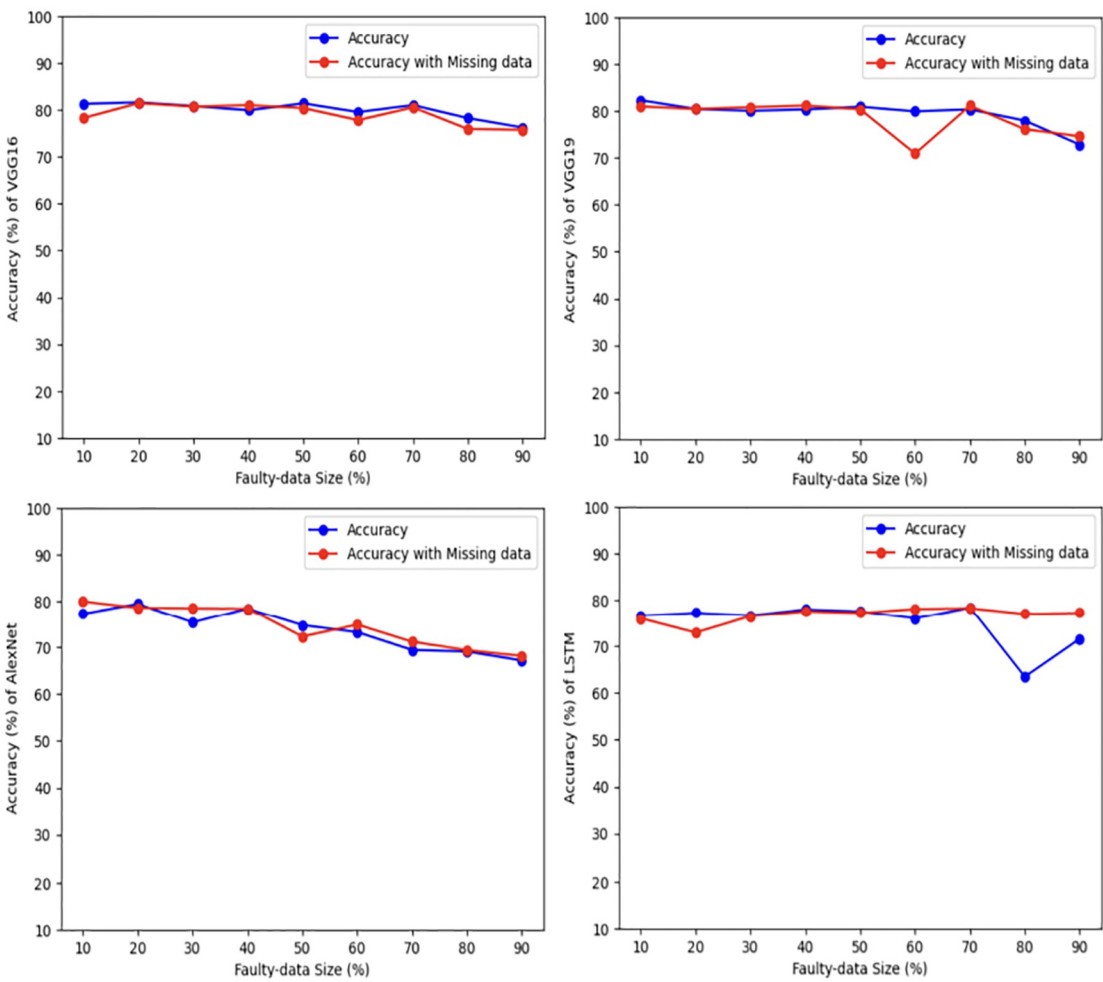

**Fig 6. Vulnerability dataset results using four deep learning models.**

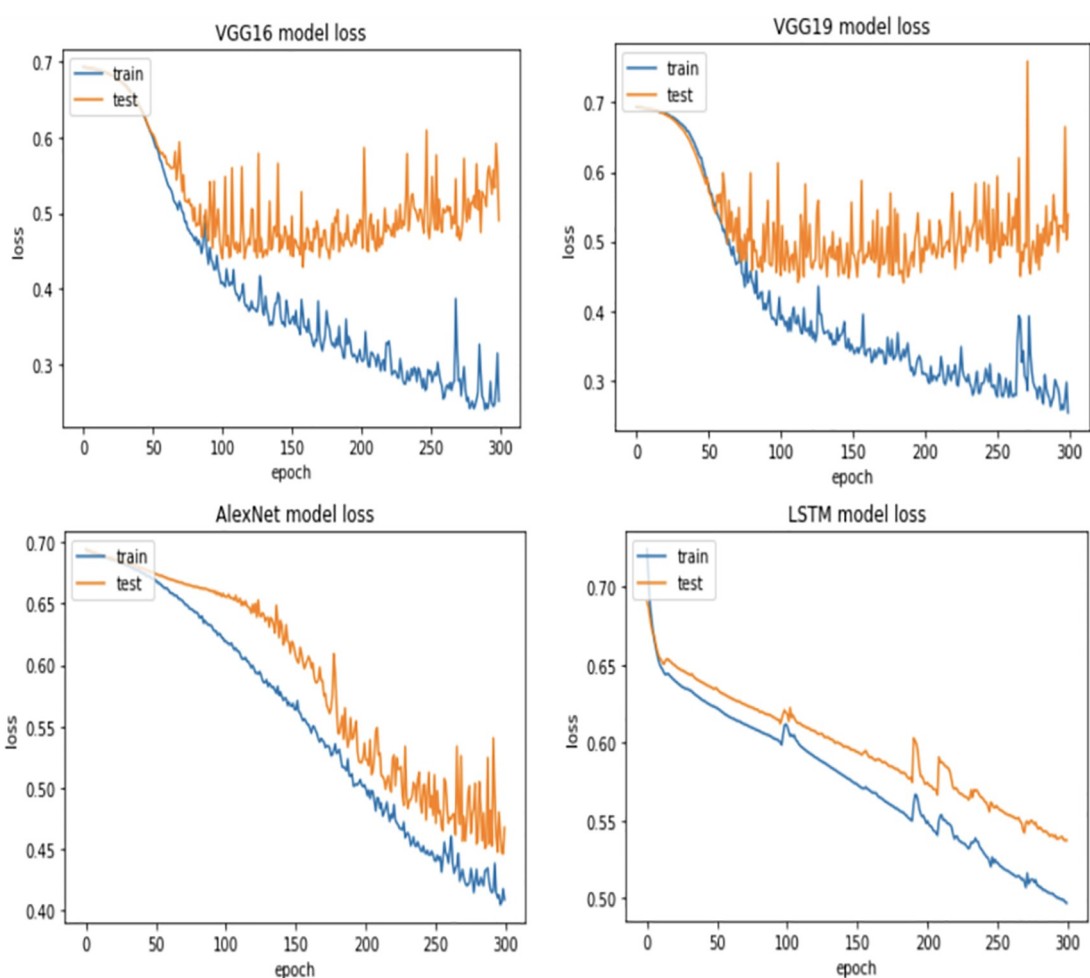

**Fig 7. Best vulnerability dataset results based on loss function.**

Furthermore, KDD Cup99 dataset is used to conduct the same tests using VGG16, VGG19, AlexNet and LSTM deep learning models with different portions of fault size. Fig 9 demonstrates KDD Cup99 dataset accuracy results using four deep learning models. As we mentioned earlier, this part focuses only on the first two scenarios. The accuracy of both scenarios is almost the same for all models. Indeed, VGG16, VGG19, and AlexNet outperform LSTM under all sizes of faulty data. In addition, Fig 10 shows the best KDD Cup99 dataset results based on the loss function. In which, the experiment shows that the loss function decreases as the faulty data size decreases and the number of epochs increases with all models. Moreover, Fig 11 illustrates the best missing intrusion dataset results based on the loss function, where the loss function decreases and the number of epochs increases as the faulty data size increases with all models.

## 6 Limitations and threat of validity

Now, it is important to illustrate the challenges and the threat of validity of the proposed model, as follows:

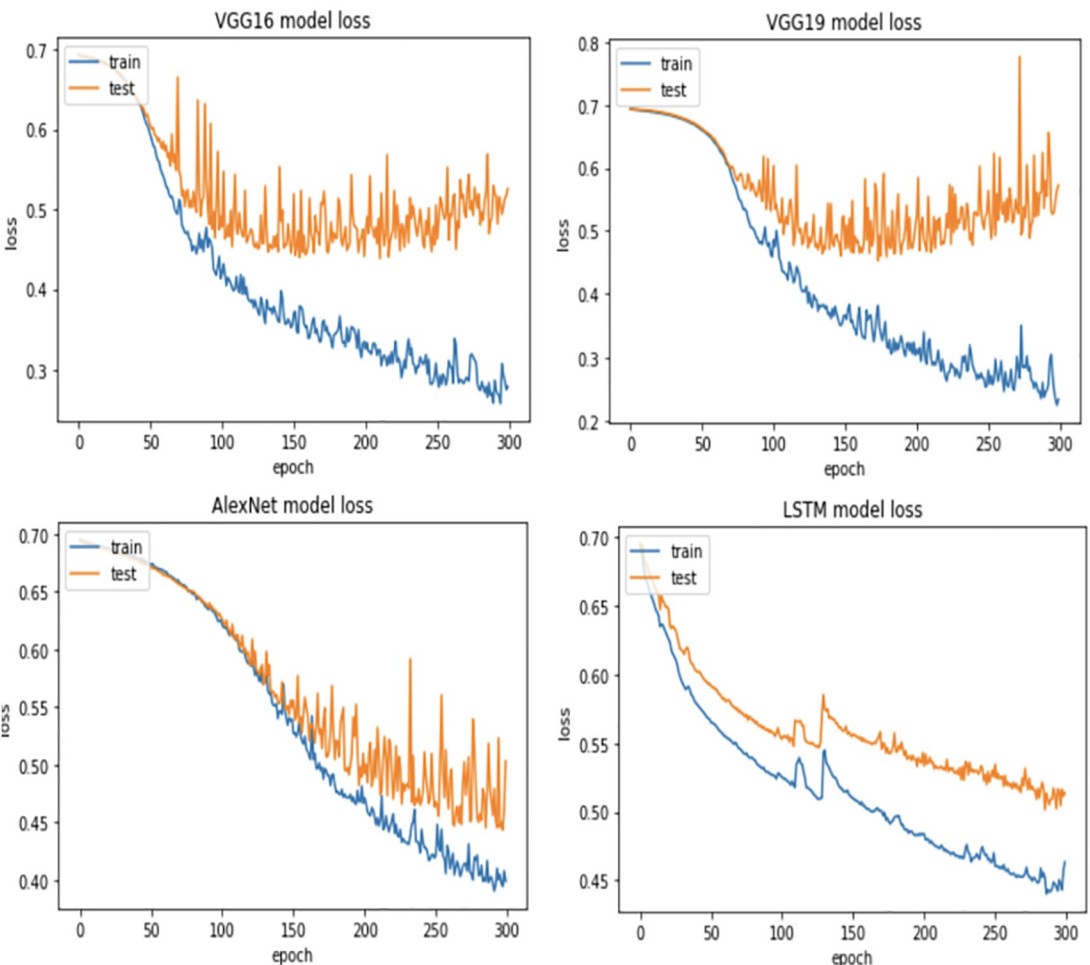

**Fig 8. Best missing vulnerability dataset results based on loss function.**

- Fault Tolerance Scope: Employing intelligent techniques such as deep learning techniques in the distributed system and fault recovery is an innovative task, without deep history or approved benchmark. Actually, deep learning models are effective in managing certain types of faults within distributed systems. However, their approach may not cover all potential fault scenarios, especially those involving complex, interconnected faults that can propagate across various system components.

- Dataset Dependence: Finding an existing suitable dataset to test the proposed idea is another challenge. In fact, the models' performance is heavily reliant on the quality and nature of the datasets they are trained on. For example, models trained on specific structured or unstructured data types might not perform optimally when applied to different data types. Therefore, train our models on both structured and unstructured datasets.

- Fault Types: Examining different kinds of faults is another issue, where we investigate different kinds of faults, using three scenarios.

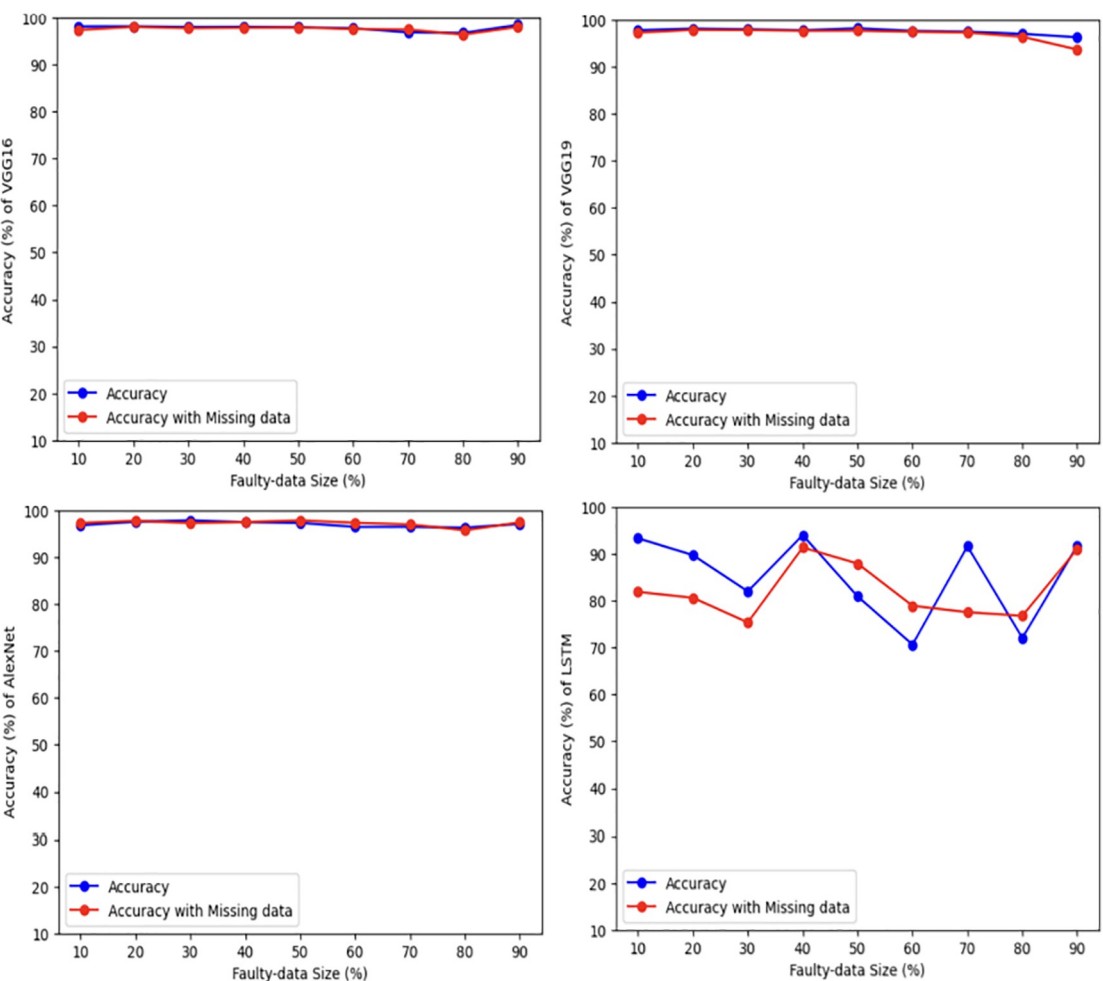

**Fig 9. KDD Cup99 dataset results using four deep learning models.**

- Fault Ratio: Fault ratios are another critical issue to investigate. Our work extends the experiment to test different fault ratios. Moreover, in cases involving larger faults, the recovery time could be considerable.

- Computational Overhead: Implementing deep learning models for real-time fault detection and correction introduces computational overhead, which could be problematic for systems with stringent latency requirements. For example, the time taken for a model to make predictions during the fault recovery process can also be a bottleneck, especially in real-time systems where rapid decision-making is essential.

- Training Time: Training deep learning models, particularly with large datasets or complex architectures like VGG16, VGG19, or ResNet34, is time-intensive. This significant time investment must be taken into account when deploying these models in practical applications.

- Additionally, finding proper deep learning techniques and to generalize our findings are vital points to consider.

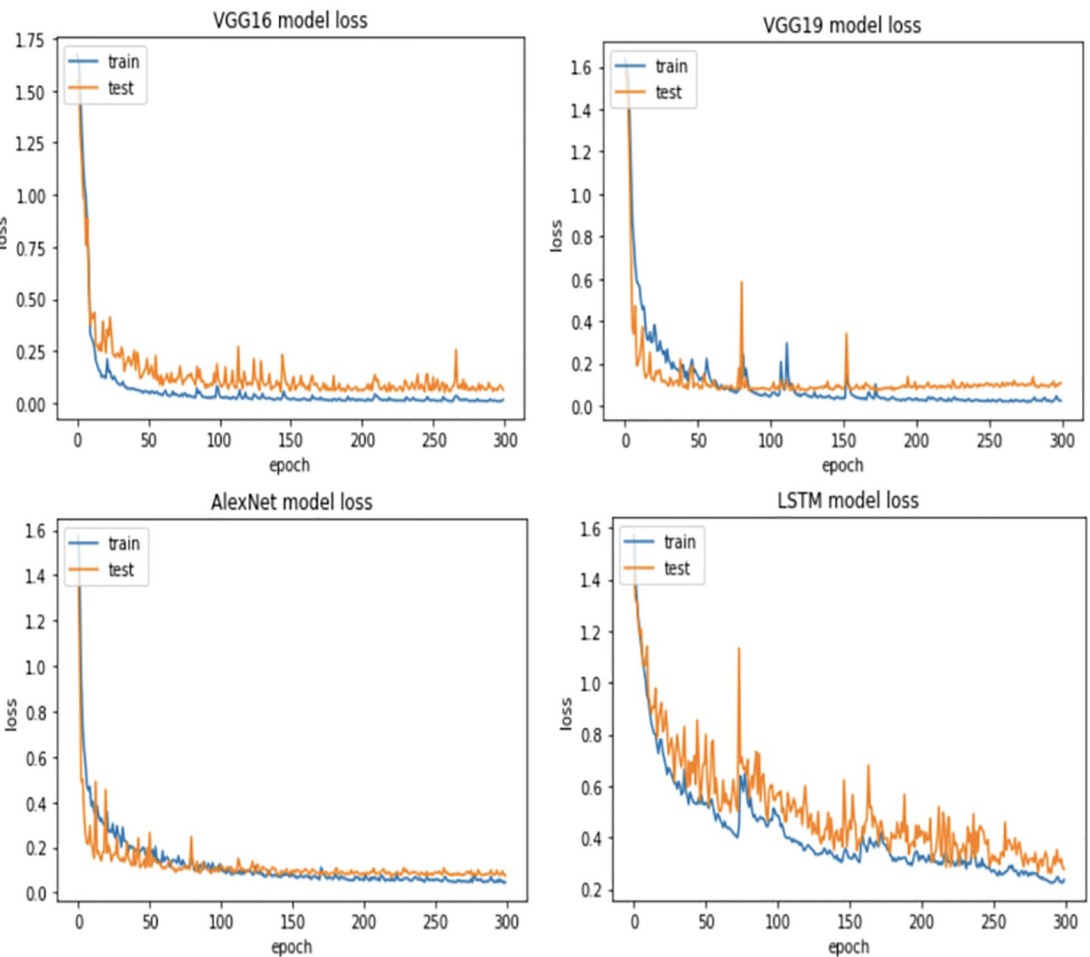

**Fig 10. Best KDD Cup99 dataset results based on loss function.**

## 7 Conclusion

This work leverages the strengths of deep learning, particularly in pattern recognition and prediction, to address errors in distributed systems across three scenarios:

- Processor Malfunction: When a processor fails to generate the correct output, deep learning models can use the inputs and outputs from other functioning processors to identify patterns and reconstruct the correct output of the malfunctioning processor.

- Corrupted Inputs: The deep learning model learns from the inputs and outputs of successful processors to detect patterns and accurately predict the correct output of faulty processors, even when their inputs are corrupted.

- Distinct Input Patterns: In cases where the input patterns of unsuccessful processors differ significantly from those of successful ones, the deep learning model can identify these as new patterns, categorizing them as *Unknown*.

We employ deep learning architectures such as VGG16, VGG19, AlexNet, LSTM, and ResNet34 to evaluate the performance of deep learning in these scenarios. The analysis

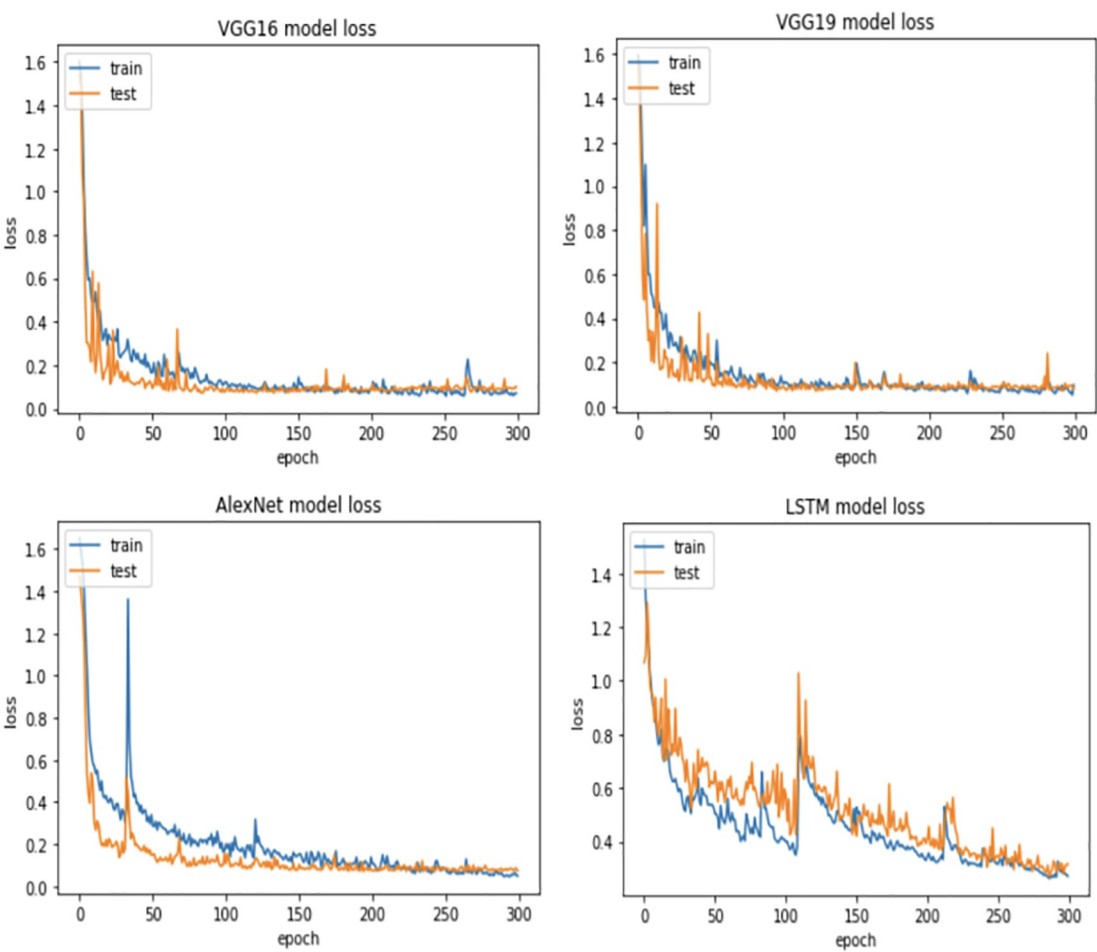

**Fig 11. Best missing KDD Cup99 dataset results based on loss function.**

includes both structured and unstructured data, demonstrating that our models achieve high detection accuracy when compared to non-faulty processing and data.

## Author Contributions

**Conceptualization:** Basem Assiri.

**Data curation:** Abdullah Sheneamer.

**Formal analysis:** Basem Assiri.

**Funding acquisition:** Abdullah Sheneamer.

**Investigation:** Basem Assiri, Abdullah Sheneamer.

**Methodology:** Basem Assiri.

**Project administration:** Basem Assiri.

**Resources:** Basem Assiri.

**Validation:** Basem Assiri.

**Visualization:** Abdullah Sheneamer.

**Writing – original draft:** Basem Assiri, Abdullah Sheneamer.

**Writing – review & editing:** Basem Assiri.

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
