## [Editor Report · Decision Letter 0]

26 Mar 2024

PONE-D-24-09708Fault Tolerance in Distributed Systems Using Deep Learning ApproachesPLOS ONE

Dear Dr. Sheneamer,

Thank you for submitting your manuscript to PLOS ONE. After careful consideration, we feel that it has merit but does not fully meet PLOS ONE’s publication criteria as it currently stands. Therefore, we invite you to submit a revised version of the manuscript that addresses the points raised during the review process.

We look forward to receiving your revised manuscript.

Kind regards,

Abul Bashar

Academic Editor

PLOS ONE

 [The authors extend the appreciation to the Deputyship for Research Innovation, Ministry of Education in Saudi Arabia for funding this research work through project number:ISP-2024]. 

4. Please update your submission to use the PLOS LaTeX template. The template and more information on our requirements for LaTeX submissions can be found at http://journals.plos.org/plosone/s/latex.

5.  We note that your Data Availability Statement is currently as follows: [We use some standardized datasets and they are cited]

6. Please amend your list of authors on the manuscript to ensure that each author is linked to an affiliation. Authors’ affiliations should reflect the institution where the work was done (if authors moved subsequently, you can also list the new affiliation stating “current affiliation:….” as necessary).

7. We note that Figure(s) 2 and 3 in your submission contain copyrighted images. All PLOS content is published under the Creative Commons Attribution License (CC BY 4.0), which means that the manuscript, images, and Supporting Information files will be freely available online, and any third party is permitted to access, download, copy, distribute, and use these materials in any way, even commercially, with proper attribution. For more information, see our copyright guidelines: http://journals.plos.org/plosone/s/licenses-and-copyright.

A. You may seek permission from the original copyright holder of Figure(s) 2 and 3 to publish the content specifically under the CC BY 4.0 license.

B. If you are unable to obtain permission from the original copyright holder to publish these figures under the CC BY 4.0 license or if the copyright holder’s requirements are incompatible with the CC BY 4.0 license, please either i) remove the figure or ii) supply a replacement figure that complies with the CC BY 4.0 license. Please check copyright information on all replacement figures and update the figure caption with source information. If applicable, please specify in the figure caption text when a figure is similar but not identical to the original image and is therefore for illustrative purposes only

Additional Editor Comments 

Please submit the paper in the required PLOS ONE format.

:

---

## [Author Response · Author response to Decision Letter 0]

11 May 2024

Accept warm greetings. Thanks for considering our manuscript No. PONE-D-24-09708R1 for revision. Please find enclosed the revised manuscript titled: " Fault Tolerance in Distributed Systems Using Deep Learning Approaches. " which we are resubmitting for review and possible inclusion as an article in the coming issues of PLOS ONE

We addressed every issue raised by esteemed reviewers very carefully. Enclosed herewith responses against the issues raised by the reviewers.

This manuscript is the authors' original work and has not been published nor has it been submitted simultaneously elsewhere. 

Please address all correspondence concerning this manuscript to me. I am ready to provide any necessary supports related to the manuscript.

Best Regards,

Authors

---

## [Decision Letter · Decision Letter 1]

1 Aug 2024

PONE-D-24-09708R1Fault Tolerance in Distributed Systems Using Deep Learning ApproachesPLOS ONE

Dear Dr. Sheneamer,

Thank you for submitting your manuscript to PLOS ONE. After careful consideration, we feel that it has merit but does not fully meet PLOS ONE’s publication criteria as it currently stands. Therefore, we invite you to submit a revised version of the manuscript that addresses the points raised during the review process.

**The following UPDATED comments from the reviewer(s) are to be addressed as well.** ***The abstract is to be quantified. The highlights are not properly noted.***

**
*The limitations of previous works are to be included.  Section 2 and 3 needs improvement. Discussion part is poorly presented. Conclusion may be modified. The limitations and time loss analysis may be included. Bound condition for all equations are to be included. *
**

We look forward to receiving your revised manuscript.

Kind regards,

Abul Bashar

Academic Editor

PLOS ONE

Reviewers' comments:

Reviewer's Responses to Questions

**Comments to the Author (These comments are to be ignored, as the comments mentioned above take precedence)**

1. If the authors have adequately addressed your comments raised in a previous round of review and you feel that this manuscript is now acceptable for publication, you may indicate that here to bypass the “Comments to the Author” section, enter your conflict of interest statement in the “Confidential to Editor” section, and submit your "Accept" recommendation.

Reviewer #1: All comments have been addressed

2. Is the manuscript technically sound, and do the data support the conclusions?

Reviewer #1: Partly

3. Has the statistical analysis been performed appropriately and rigorously? 

Reviewer #1: N/A

4. Have the authors made all data underlying the findings in their manuscript fully available?

Reviewer #1: Yes

5. Is the manuscript presented in an intelligible fashion and written in standard English?

Reviewer #1: Yes

6. Review Comments to the Author

**Reviewer #1: **The abstract is to be quantified. The highlights are not properly noted. The limitations of previous works are to be included. Well corrected paper.

7. PLOS authors have the option to publish the peer review history of their article (what does this mean?). If published, this will include your full peer review and any attached files.

Reviewer #1: **Yes: **Harikumar Rajaguru

---

## [Author Response · Author response to Decision Letter 1]

16 Aug 2024

These comments below are mentioned in orgnaized table in attached repsonse to reviewers file. 

Responses to reviewer 1

First of all, we would like to thank you very much for your valuable and fruitful comments and suggestions, that are taken as a guide to improve this work.

Comment Response

The abstract is to be quantified. The highlights are not properly noted.

 Updated, the main highlights are noted, including the use of deep learning for fault tolerance and recovery in three major scenarios. 

The abstract now is quantified. The accuracies of the models are measured and the abstract shows the accuracy results. 

Thanks for your valuable comments. 

The limitations of previous works are to be included. Updated, in the related work we first highlight previous works of the distributed system works. Then, the fault tolerance concept. After that, the use of machine learning in fault tolerance and finally the use of deep learning in fault tolerance. 

At the end of first and second paragraph, in paragraph four as well, we show the limitations of the previous works and how we handle such limitation. For example, the traditional distributed system and fault tolerance do not use intelligent tools such as deep learning. In addition, they do not use deep learning for recovery, and to handle missing, corrupted and unrelated input. 

Thanks for your valuable comments. 

Section 2 and 3 needs improvement. Section 2 and 3 have been improved. The updates are highlighted. 

Thanks for your valuable comments. 

Discussion part is poorly presented. Updated, see sections 3, 4 and 6. The discussion has been improved. 

Actually, to avoid the overlength of the paper, there is no specific section of discussion. However, the discussion of the main idea is included in Section 3, while the evaluation of our model and experiment is discussed within Section 4, and the limitations and threat of validity is presented in Section 6.

Thanks for your valuable comments. 

Conclusion may be modified. The conclusion had been modified.

Thanks for your valuable comments. 

The limitations and time loss analysis may be included. Updated, now it is included in the section of Limitation and Threat of Validity 

Fault Tolerance Scope: Employing intelligent techniques such as deep learning techniques in the distributed system and fault recovery is an innovative task, without deep history or approved benchmark. Actually, deep learning models are effective in managing certain types of faults within distributed systems. However, their approach may not cover all potential fault scenarios, especially those involving complex, interconnected faults that can propagate across various system components.

Dataset Dependence: Find an existing suitable dataset to test the proposed idea is another challenge. In fact, the models’ performance is heavily reliant on the quality and nature of the datasets they are trained on. For example, models trained on specific structured or unstructured data types might not perform optimally when applied to different data types. Therefore, train our models on both structured and unstructured datasets.

Fault Types: Examine different kinds of faults is another issue, where we investigate different kinds of faults, using three scenarios.

Fault Ratio: Faults ratios is another critical issue to investigate. Our work extends the experiment to test different faults ratios. Moreover, in cases involving larger faults, the recovery time could be considerable.

Computational Overhead: Implementing deep learning models for real-time fault detection and correction introduces computational overhead, which could be problematic for systems with stringent latency requirements. For example, the time taken for a model to make predictions during the fault recovery process can also be a bottleneck, especially in real-time systems where rapid decision-making is essential.

Training Time: Training deep learning models, particularly with large datasets or complex architectures like VGG16, VGG19, or ResNet34, is time-intensive. This significant time investment must be taken into account when deploying these models in practical applications.

Additionally, finding proper deep learning techniques and to generalize our findings are vital points to consider.

Bound condition for all equations are to be included. Updated, see section 4, all equation results are between 0 and 1, as all variables are positive and the numerator is smaller than the denominator

1. If the authors have adequately addressed your comments raised in a previous round of review and you feel that this manuscript is now acceptable for publication, you may indicate that here to bypass the “Comments to the Author” section, enter your conflict of interest statement in the “Confidential to Editor” section, and submit your "Accept" recommendation.

Reviewer #1: All comments have been addressed

 Thanks for your valuable comments

2. Is the manuscript technically sound, and do the data support the conclusions?

Reviewer #1: Partly The technical part has been improved in all parts of the paper. Thanks for your valuable comments 

3. Has the statistical analysis been performed appropriately and rigorously?

Reviewer #1: N/A -

4. Have the authors made all data underlying the findings in their manuscript fully available?

Reviewer #1: Yes Thanks for your valuable comments

5. Is the manuscript presented in an intelligible fashion and written in standard English?

Reviewer #1: Yes

 Thanks for your valuable comments

---

## [Editor Report · Decision Letter 2]

5 Sep 2024

Fault Tolerance in Distributed Systems Using Deep Learning Approaches

PONE-D-24-09708R2

Dear Dr. Sheneamer,

We’re pleased to inform you that your manuscript has been judged scientifically suitable for publication and will be formally accepted for publication once it meets all outstanding technical requirements.

Kind regards,

Abul Bashar

Academic Editor

PLOS ONE
---

## [Editor Report · Acceptance letter]

26 Dec 2024

PONE-D-24-09708R2 

PLOS ONE

Dear Dr. Sheneamer, 

I'm pleased to inform you that your manuscript has been deemed suitable for publication in PLOS ONE. Congratulations! Your manuscript is now being handed over to our production team.

Kind regards, 

on behalf of

Dr. Abul Bashar 

Academic Editor

PLOS ONE